# A Global Analysis of Y-STR *INRA189* Polymorphism in Chinese Domestic Yak Breeds/Populations

**DOI:** 10.3390/ani10030393

**Published:** 2020-02-28

**Authors:** Zhijie Ma, Xiaoting Xia, Shengmei Chen, Ma Bai, Chuzhao Lei, Jianlin Han

**Affiliations:** 1Qinghai Academy of Animal Science and Veterinary Medicine, Qinghai University, Xining 810016, China; c504394191@163.com; 2College of Animal Science and Technology, Northwest A&F University, Yangling 712100, China; xiaxiaoting1991@163.com (X.X.); leichuzhao1118@126.com (C.L.); 3Bureau of Animal Husbandry and Veterinary, Diqing Tibetan Autonomous Prefecture of Yunnan Province, Shangri-La 674400, China; 13988733843@sina.cn; 4CAAS-ILRI Joint Laboratory on Livestock and Forage Genetic Resources, Institute of Animal Science, Chinese Academy of Agricultural Sciences (CAAS), Beijing 100193, China; h.jianlin@cgiar.org; 5International Livestock Research Institute (ILRI), Nairobi 00100, Kenya

**Keywords:** *Bos grunniens*, Y-specific microsatellite, *INRA189*, polymorphism, repeat motif, geographical distribution pattern

## Abstract

**Simple Summary:**

Y chromosome-specific markers have been widely used in studying the origin, migration, diversity, and population relationship in several mammalian species. So far, the investigations on yak paternal genetics were mostly based on Y-SNPs information, but very few on Y-STR markers. In this study, we comprehensively investigated the polymorphism of Y-STR *INRA189* locus, identified the geographical distribution pattern of its alleles in 15 Chinese domestic yak breeds/populations, and provided basic data for yak paternal genetic analysis. Our results showed that *INRA189* is an intermediate polymorphic Y-STR marker with six alleles present among 15 Chinese domestic yak breeds/populations and that the variation among six alleles was due to different numbers of repetitions of a TG dinucleotide motif.

**Abstract:**

The objective of this study was to probe into the polymorphism of Y-STR *INRA189* and identify the geographical distribution pattern of its alleles in the Chinese domestic yak gene pool. We examined the variation at *INRA189* locus in 682 male yaks representing 15 breeds/populations in China. The results showed that six alleles, including five reported previously (149, 155, 157, 159, and 161 bp) and a new one identified in this study (139 bp), were detected at *INRA189* locus based on genotyping analysis. The frequencies of six alleles varied among the 15 yak breeds/populations with a clear phylogeographical pattern, which revealed the paternal genetic difference among Chinese yak breeds/populations. The average polymorphism information content (PIC) among the 15 yak breeds/populations was 0.32, indicating *INRA189* to be an intermediate polymorphic Y-STR marker (0.25 < PIC < 0.5) in yak. Sequence alignment revealed that the variations among six alleles at *INRA189* were defined by a TG dinucleotide repeat motif, which repeated for 12, 17, 20, 21, 22, and 23 times, corresponding to the alleles of 139, 149, 155, 157, 159, and 161 bp, respectively. Therefore, we believe that the polymorphic yak Y-STR *INRA189* can be used to characterize male-mediated genetic events, including paternal genetic origin, diversity, and evolution.

## 1. Introduction

Yak (*Bos grunniens*) is one of the native mammalian species that lives on the Qinghai-Tibetan Plateau and its adjacent territories. They provide products (milk, meat, hair, hides, and manure for fuel) and services (draught, packing, and riding) to pastoralists and agro-pastoralists living in these areas [1]. China owns rich yak genetic resources with more than 15 million animals represented by 12 officially recognized breeds and several local populations [2].

Y-chromosome-specific polymorphisms (such as single nucleotide polymorphisms (SNPs), insertion-deletion mutations (Indels), microsatellites with short tandem repeats (STRs) and copy number variations (CNVs)) can add significant genetic information to what can be inferred from mtDNA and autosomal variations [3,4]. Thus far, five Y-SNP markers have been used to explore the origin, diversity, and phylogeographical structure of the Chinese domestic yak [5,6,7], whereas a few Y-chromosome-specific microsatellite (Y-STR) markers were identified in yak [8]. Collectively, several previous studies focused on the specificity and polymorphism of *INRA189* locus in yak. For example, Han et al. (2000) identified the male specificity and detected three alleles at *INRA189* locus in one Bhutanese yak population and three Chinese yak breeds (Gannan, Tianzhu, and Datong) [9]. Nguyen et al. (2005) found two alleles at this locus in one Swiss yak population [10]. In addition, our analysis on *INRA189* in three Chinese yak breeds (Gaoyuan, Huanhu, and Datong) indicated that this marker possessed four alleles [11]. It can be seen from the above-mentioned studies that only a few Chinese domestic yak breeds/populations were tested for *INRA189* polymorphism, whereas no comprehensive study was reported in all Chinese yak breeds/populations. Under this scenario, the aim of this study was to further evaluate the polymorphism and identify the geographical distribution pattern of different alleles at Y-STR *INRA189* locus in 15 domestic yak breeds/populations in China.

## 2. Materials and Methods

Blood or tissue samples were collected from 682 male yaks across the distribution ranges of 15 breeds/populations in China, namely Gaoyuan, Datong, Huanhu, Xueduo, Tianzhu, Gannan, Jiulong, Maiwa, Sibu, Niangya, Pali, Leiwuqi, Bazhou, Taxian, and Zhongdian (Figure 1, Table 1). Five females of the Gaoyuan breed were selected as controls. These yaks were selected randomly based on pedigree information or detailed interviews with owners to minimize the degree of relationship among individuals. The study was approved by the Faculty of Animal Policy and Welfare Committee of Qinghai University.

Genomic DNA was extracted using the Aidlab genomic DNA extraction kit (Aidlab Biotechnologies Co., Ltd., China). A pair of primers (PF: 5′-TTTTGTTTCCCGTGCTGAG -3′; PR: 5′-GAACCTCGTCTCCTTGTAGCC-3′) described by Kappes et al. (1997) was used to amplify the *INRA189* locus [12] following the protocols for PCR amplification and genotyping analysis reported in our previous study [11].

Purified PCR products were scanned together with the GS-500 Liz size standard in ABI 3730XL Sequencer (Applied Biosystems, Foster City, CA, USA). Allele sizes were determined for each individual using GeneMarker v1.97 (http://www.softgenetics.com/GeneMarker.html) and the allele frequencies were estimated using the GENETIX 4.05 software [13]. In order to obtain the informativeness of the *INRA 189* marker, the polymorphic information content (PIC) was calculated using PIC_CALC 0.1 software [14]. To confirm the sequence characterization of each allele by DNA sequencing, we cloned PCR amplicons from different alleles into the pUCm-T vector (Sangon Biotech Co., Ltd., Shanghai, China) and carried out sequence alignment using BioEdit v7.2.5 software with Clustal W multiple alignment [15].

## 3. Results and Discussion

In accordance with previous reports [9,10,11], our current results also showed that *INRA189* was only present in male yaks, which further confirmed it to be a yak Y-chromosome specific microsatellite marker. In analysis of yak *INRA189* polymorphism, previous studies have identified five alleles (149, 155, 157, 159, and 161 bp) at *INRA189* locus based on the genotyping analysis of one Swiss population, one Bhutanese population and five Chinese breeds (Gannan, Tianzhu, Gaoyuan, Huanhu, and Datong) [9,10,11]. Herein, we comprehensively examined the polymorphism of *INRA189* marker in 15 Chinese yak breeds/populations. Our results showed that six alleles, including those five reported previously [9,10,11] and a new one identified in this study (139 bp), were detected in 15 yak breeds/populations (Figure 2, Table 1). The frequencies of six alleles varied among 15 yak breeds/populations (Figure 1, Table 1). The 157 bp allele was predominant and shared among all yak breeds/populations with frequencies ranging between 6.1% in Pali and 100% in Leiwuqi and Zhongdian, followed by the 155 bp allele present in 13 breeds/populations with frequencies varying from 2.9% in Gannan and Maiwa to 93.9% in Pali. The 159 bp allele was only present in five breeds/populations (Datong, Huanhu, Xueduo, Maiwa, and Bazhou) at an overall frequency of 1.6% while the 161 bp allele was only found in Gaoyuan, Tianzhu, and Taxian breeds/populations at 3.1%. Interestingly, private 139 bp and 149 bp alleles were only detected in Gaoyuan breed with frequencies of 0.5% and 1.4%, respectively. The above results showed that 155 and 157 bp alleles were the dominant alleles, but others including 139, 149, 159, and 161 bp alleles were rare alleles in domestic yak. Generally, dominant alleles reflect basic breed/population characteristics while rare alleles display breed/population uniqueness of great value in breeding practice. Therefore, Gaoyuan, Datong, Huanhu, Xueduo, Tianzhu, Maiwa, Bazhou, and Taxian breeds/populations should be paid more attention in their future conservation and breeding programs as they owned rare alleles. Furthermore, Pali breed could be regarded as a unique genetic resource because it had a higher frequency of 155 bp allele, which was different from other yak breeds/populations. It is evident that genetic drift, natural selection, and artificial breeding may have played a role in driving the differences in the number and frequency of alleles at *INRA189* locus among the 15 yak breeds/populations. In addition, based on the autosomal microsatellites [16], mtDNA [17], and Y-SNP markers [7], previous studies have explored the genetic differences among some Chinese yak breeds/populations. Here, although the information from *INRA189* marker was limited, our results reflected the genetic differences among these yak breeds/populations to some extent (Figure 1, Table 1). Even though our results are not completely consistent with those of previous studies [7,16,17], *INRA189* polymorphism among these domestic yak breeds/populations will add paternal genetic information to what can be inferred from autosomal microsatellites, mtDNA, and Y-SNP markers. Therefore, we believe that the Y-STR *INRA189* combining with other molecular markers (e.g., Y-SNPs) can be used to characterize male-mediated genetic events in yak, including paternal origin, evolution, diversity, and population structure.

The polymorphism information content (PIC) provides an estimate of the discriminatory power of a locus. The value of PIC reflects the level of polymorphism of microsatellite markers in animal populations. According to the classification standard of Botstein et al. (1980) [18], markers with PIC values greater than 0.5 are considered to be very informative, values between 0.25 and 0.50 are moderately informative while values lower than 0.25 are minimally informative. In our previous study, the results showed that the average PIC for *INRA189* marker in three Chinese yak breeds (Gaoyuan, Huanhu, and Datong) was 0.2379, which indicated that this marker was a minimally informative STR marker in yak (PIC < 0.25). However, in this study, the average PIC value among the 15 domestic yak breeds/populations was 0.32, indicating *INRA189* to be an intermediately polymorphic Y-STR marker (0.25 < PIC < 0.5). This result changed our previous view on the polymorphism of this marker in yak [11].

The result of sequence alignment in this study revealed the variation among the six alleles defined by a TG dinucleotide repeat motif, which repeated for 12, 17, 20, 21, 22, and 23 times corresponding to the alleles of 139, 149, 155, 157, 159, and 161 bp, respectively (Figure 3). Our result not only identified the sequence of the Y-STR *INRA189* locus, but also clarified the phylogeographic pattern among the six alleles in domestic yak.

## 4. Conclusions

In conclusion, our results showed that *INRA189* locus is an intermediate polymorphic Y-chromosome-specific microsatellite marker in yak. We identified six alleles (139, 149, 155, 157, 159, and 161 bp) with varying frequencies among 15 Chinese yak breeds/populations that followed a clear phylogeographical pattern, indicating paternal genetic differences among these yak breeds/populations. The variations among the six alleles were due to a TG dinucleotide motif.

## Figures and Tables

**Figure 1 animals-10-00393-f001:**
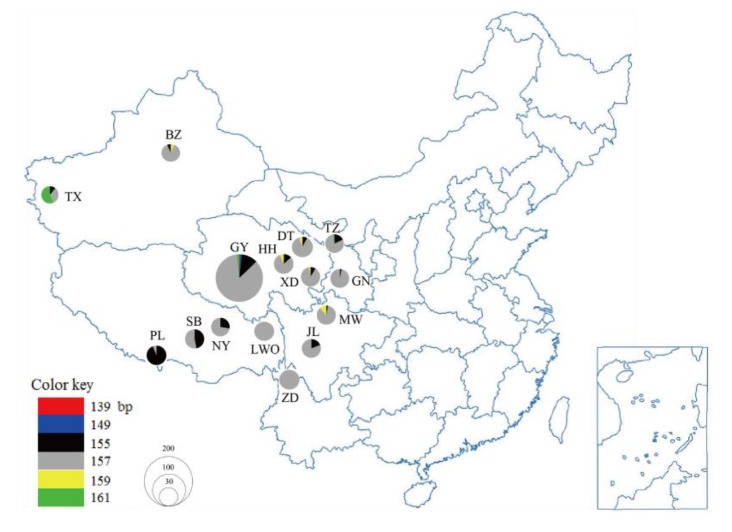
Geographical distribution of six alleles at Y-STR *INRA189* locus in 15 Chinese yak breeds/populations. Circle area is proportional to the sample size. GY, Gaoyuan; DT, Datong; HH, Huanhu; XD, Xueduo; TZ, Tianzhu; GN, Gannan; JL, Jiulong; MW, Maiwa; SB, Sibu; NY, Niangya; PL, Pali; LWQ, Leiwuqi; BZ, Bazhou; TX, Taxian, and ZD, Zhongdian.

**Figure 2 animals-10-00393-f002:**
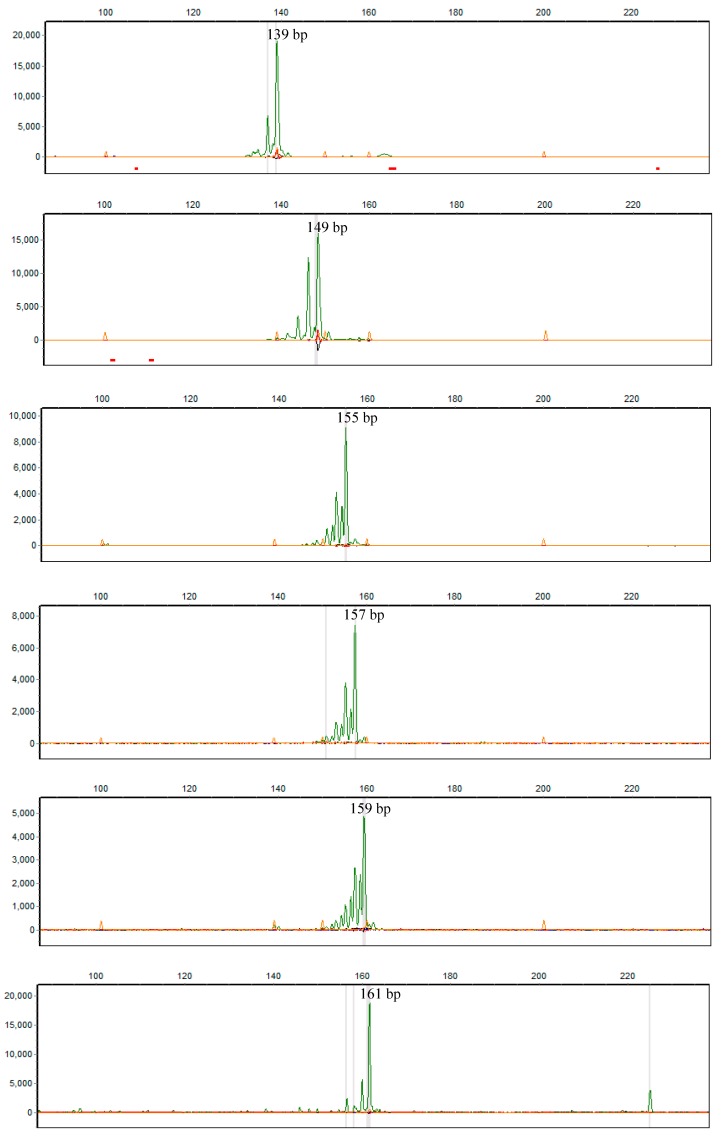
Genotyping of yak Y-STR *INRA189* marker.

**Figure 3 animals-10-00393-f003:**
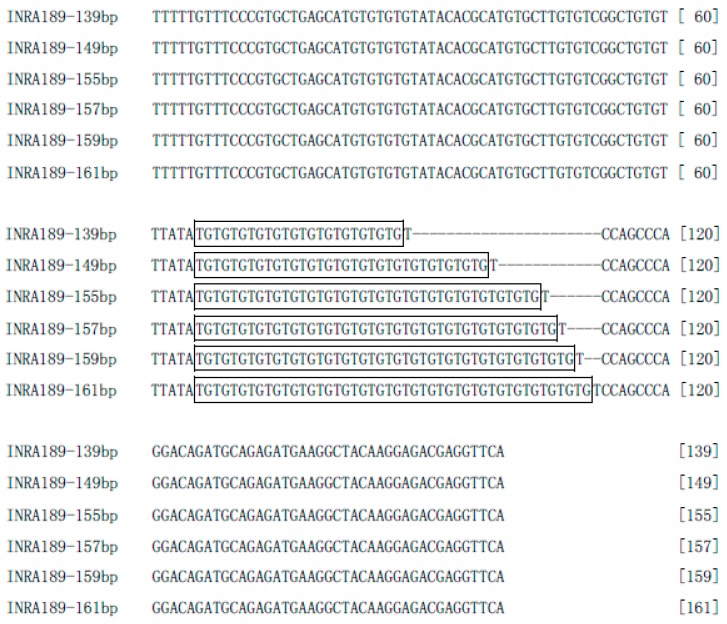
Sequence alignment of six alleles at yak Y-STR *INRA189* locus. Dinucleotide repeat motif is marked by square box. Horizontal lines (-) denote nucleotide InDel sites.

**Table 1 animals-10-00393-t001:** Allele frequencies at *INRA189* locus in 15 Chinese yak breeds/populations.

Province/Region	Breed/Population (Code)	Sampling Location	Sample No.	Different Alleles at *INRA189* Locus (bp)					
				139	149	155	157	159	161
Qinghai	Gaoyuan (GY)	Qumalai, Tianjun, Qilian, Gande and Henan counties; Tanggulashan and Guoleimude towns	218	0.005 (1)	0.014 (3)	0.010 (24)	0.853 (186)		0.018 (4)
	Datong (DT)	Datong yak breeding farm	40			0.075 (3)	0.875 (35)	0.050 (2)	
	Huanhu (HH)	Gonghe county	30			0.133 (4)	0.800 (24)	0.067 (2)	
	Xueduo (XD)	Henan county	35			0.086 (3)	0.886 (31)	0.028 (1)	
Gansu	Tianzhu (TZ)	Tianzhu county	34			0.177 (6)	0.794 (27)		0.029 (1)
	Gannan (GN)	Luqu county	34			0.029 (1)	0.971 (33)		
Sichuan	Jiulong (JL)	Jiulong county	32			0.188 (6)	0.812 (26)		
	Maiwa (MW)	Hongyuan county	35			0.029 (1)	0.857 (30)	0.114 (4)	
Tibet	Sibu (SB)	Mozhugongka county	34			0.471 (16)	0.529 (18)		
	Niangya (NY)	Jiali county	33			0.273 (9)	0.727 (24)		
	Pali (PL)	Yadong county	33			0.939 (31)	0.061 (2)		
	Leiwuqi (LWQ)	Leiwuqi county	29				1.000(29)		
Xinjiang	Bazhou (BZ)	Hejing county	33			0.061 (2)	0.878 (29)	0.061 (2)	
	Taxian (TX)	Tajik autonomous county of Taxkorgan	28			0.061 (3)	0.878 (9)		0.061 (16)
Yunnan	Zhongdian (ZD)	Shangri-la city	34				1.000 (34)		
Total			682	0.002 (1)	0.004 (3)	0.160 (109)	0.787 (537)	0.016 (11)	0.031 (21)

Note: Numbers in parentheses represent the numbers of animals carrying each allele.

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
