# Peer review of "A Global Analysis of Y-STR INRA189 Polymorphism in Chinese Domestic Yak Breeds/Populations"

_animals, 2020, doi:10.3390/ani10030393_

Round 1

Reviewer 1 Report

The paper extends and improves the investigation on the Y INRA189 STR marker in Chinese yak breeds/populations (see reference 11). The paper lacks of substantial amount of new information about this topic, nevertheless it may be accepted with some modifications.

Comments.

The INRA189 repeat unit is TG, not TA, if figure 3 is considered.

Line 4. Perhaps a seventh author was left out, otherwise erase 'and 5' and the affiliation 5.

Lines 23-24. Replace 'variations' with 'variation'. Replace 'were accused by repeat time of the TA' with 'was due to number of repetitions of the TG'.

Line 25. Add 'The objective of this investigation was to probe'.

Line 30. Add 'the' before '15'.

Line 35. Replace 'TA' with 'TG'.

Line 48. Erase 'number of' and add 'head' after 'million'.

Line 54. Replace '. However' with ', whereas'.

Line 62. Replace '. No' with ', whereas no'.

Line 63. Erase 'on INRA189 marker'.

Line 85. Erase 'at the same'.

Line 96. Erase 'in this study'.

Line 97. Erase 'INRA189'.

Line 104. Erase 'the INRA189 genotyping analysis of'.

Line 110. Replace 'at 1.6%' with 'at an overall frequency of'.

Lines 130-132. Peraps it would be useful to support genetic difference among breeds/populations using a short comparison with the results of other investigations (see reference 7 and Zhang et al. (2008) Journal of Genetics and Genomics 35, 233).

Line 201. Replace 'accused by the repeat time of TA' with 'due to a TG'.

Author Response

Point 1: The INRA189 repeat unit is TG, not TA, if figure 3 is considered.

Response 1:“TA”has been changed to “TG”in the full manuscript. Sorry for our carelessness. Thanks! (see page 1, line 24 and 35; page 5, line 150 and 159).

Point 2: Line 4. Perhaps a seventh author was left out, otherwise erase 'and 5' and the affiliation 5.

Response 2: Have done. Thanks (see page 1, line 4).

Point 3: Lines 23-24. Replace 'variations' with 'variation'. Replace 'were accused by repeat time of the TA' with 'was due to number of repetitions of the TG'.

Response 3: Have done. Thanks (see page 1, line 23-24).

Point 4: Line 25. Add 'The objective of this investigation was to probe'.

Response 4: Have done. Thanks (see page 1, line 25).

Point 5: Line 30. Add 'the' before '15'.

Response 5: Have done. Thanks (see page 1, line 30).

Point 6: Line 35. Replace 'TA' with 'TG'.

Response 4: Have done. Thanks (see page 1, line 35).

Point 7: Line 48. Erase 'number of' and add 'head' after 'million'.

Response 7: Have done. Thanks (see page 2, line 48).

Point 8: Line 54. Replace '. However' with ', whereas'.

Response 8: Have done. Thanks (see page 2, line 54).

Point 9: Line 62. Replace '. No' with ', whereas no'.

Response 9: Have done. Thanks (see page 2, line 63).

Point 10: Line 63. Erase 'on INRA189 marker'.

Response 10: Have done. Thanks (see page 2, line 63).

Point 11: Line 85. Erase 'at the same'.

Response 11: Have done. Thanks (see page 4, line 85).

Point 12: Line 96. Erase 'in this study'.

Response 12: Have done. Thanks (see page 4, line 97).

Point 13: Line 97. Erase 'INRA189'.

Response 13: Have done. Thanks (see page 4, line 98).

Point 14: Line 104. Erase 'the INRA189 genotyping analysis of'.

Response 14: Have done. Thanks (see page 4, line 104).

Point 15: Line 110. Replace 'at 1.6%' with 'at an overall frequency of'.

Response 15: Have done. Thanks (see page 4, line 110).

Point 16: Lines 130-132. Peraps it would be useful to support genetic difference among breeds/populations using a short comparison with the results of other investigations (see reference 7 and Zhang et al. (2008) Journal of Genetics and Genomics 35, 233).

Response 16: According to the reviewer's suggestion, we have added the information from previous studies including mtDNA, autosomal microsatellites and Y-SNPs analysis, and compared the investigations to support genetic difference among breeds/populations (see page 4 and 5, line 122-137). Meanwhile, we added the corresponding references (see page 8, line 288-293).

Point 17: Line 201. Replace 'accused by the repeat time of TA' with 'due to a TG'.

Response 17: Have done. Thanks (see page 5, line 159).

Reviewer 2 Report

The authors in their manuscript undertook to characterize the Y-STR INRA189 polymorphism in 15 yak breeds / populations in China. As a result of the conducted research, they found that there is geographical diversity in the distribution of individual variants of the Y-STR INRA189 marker. The work is cognitive, quite interesting, but not very innovative. It is a pity that the authors did not decide on a phylogenetic analysis of the results obtained, which would increase the value of the work.

Minor comments:

The manuscript text item 4 is missing from the reference list

Line 84 and line 85 no italics for INRA189.

Author Response

Point 1: The manuscript text item 4 is missing from the reference list.

Response 1: Have done. Thanks (see page 2, line 53).

Point 2: Line 84 and line 85 no italics for INRA189.

Response 2: Have done. Thanks (see page 4, line 85).

Reviewer 3 Report

I have some comments/suggestions to the manuscript described below.

1) P1L26: [~~ domestic yak. We examined ~~] may be [~~ domestic yak, we examined ~~].

2) P2L70-71: Authors need to explain why it is reasonable that five females of Gaoyuan breed were selected as controls in this study.

3) P4L105-108: I think it is better to discuss why allele frequencies were different (157 bp allele was so predominant), especially in terms of breeding scheme and/or management style. This discussion would give further meaningful insight into the results.

Author Response

Point 1: P1L26: [~~ domestic yak. We examined ~~] may be [~~ domestic yak, we examined ~~].

Response 1: Have done. Thanks (see page 1, line 25-27).

Point 2: P2L70-71: Authors need to explain why it is reasonable that five females of Gaoyuan breed were selected as controls in this study.

Response 2: Although INRA189 is a Y-specific microsatellite marker in yak, we still selected female yaks as control in our experiment to further identify its male specificity. Generally, in such an experiment, we need to select at least three females as control. Therefore, we think that it is reasonable that five females of Gaoyuan breed were selected as controls in this study.

Point 3: P4L105-108: I think it is better to discuss why allele frequencies were different (157bp allele was so predominant), especially in terms of breeding scheme and/or management style. This discussion would give further meaningful insight into the results.

Response 3: According to the reviewer's suggestion, we discussed the reason which caused the difference of allele frequencies (see page 4, line 113-122).
